# Accurate quantification of the stability of the perylene-tetracarboxylic dianhydride on Au(111) molecule–surface interface

Victor G. Ruiz [1✉], Christian Wagner [2,3], Friedrich Maaß[4], Hadi H. Arefi [2,3], Stephan Stremlau[4], Petra Tegeder [4], F. Stefan Tautz [2,3,5] & Alexandre Tkatchenko [6✉]

Studying inorganic/organic hybrid systems is a stepping stone towards the design of increasingly complex interfaces. A predictive understanding requires robust experimental and theoretical tools to foster trust in the obtained results. The adsorption energy is particularly challenging in this respect, since experimental methods are scarce and the results have large uncertainties even for the most widely studied systems. Here we combine temperature-programmed desorption (TPD), single-molecule atomic force microscopy (AFM), and non-local density-functional theory (DFT) calculations, to accurately characterize the stability of a widely studied interface consisting of perylene-tetracarboxylic dianhydride (PTCDA) molecules on Au(111). This network of methods lets us firmly establish the adsorption energy of PTCDA/Au(111) via TPD (1.74 ± 0.10 eV) and single-molecule AFM (2.00 ± 0.25 eV) experiments which agree within error bars, exemplifying how implicit replicability in a research design can benefit the investigation of complex materials properties.

[1] Helmholtz-Zentrum Berlin für Materialien und Energie, Hahn-Meitner-Platz 1, 14109 Berlin, Germany. [2] Peter Grünberg Institut, Forschungszentrum Jülich, 52425 Jülich, Germany. [3] Jülich Aachen Research Alliance (JARA)-Fundamentals of Future Information Technology, 52425 Jülich, Germany. [4] Ruprecht-Karls-Universität Heidelberg, Physikalisch-Chemisches Institut, Im Neuenheimer Feld 253, 69120 Heidelberg, Germany. [5] Experimentalphysik IV A, RWTH Aachen University, Otto-Blumenthal-Straße, 52074 Aachen, Germany. [6] Department of Physics and Materials Science, University of Luxembourg, L-1511 Luxembourg City, Luxembourg. ✉email: vicruiz85@gmail.com; alexandre.tkatchenko@uni.lu

Surface and interface science is an interdisciplinary research field at the origin of foundational innovations[1–3]. The last two decades have seen the emergence of a new stage in which atomic-level interface science has expanded its scope to systems of increasing complexity[1,4] including biological materials[5–8], solid-liquid interfaces[9–12], and multifunctional adsorption systems[13–20]. In particular, interfaces formed upon the adsorption of functional organic molecules on inorganic surfaces are playing a prominent role due to their implications in fundamental science[20–30] and technological applications[19,20,31–33]. The establishment of quantitative benchmarks for hybrid inorganic/organic systems (HIOS) is crucial to understand and predict their structure, stability, and dynamics[34]. However, such benchmarks are scarce, due to the shear complexity of attaining atomic-level control in complex systems and a manifold of collective electronic and atomic interactions, whose description requires the highest levels of quantum and statistical mechanics.

In this context, refinement of experimental techniques such as the normal incidence x-ray standing wave (NIXSW) and atomic force microscopy (AFM) have led to the accurate determination of adsorption geometries of HIOS irrespective of the size of the organic adsorbate[35–42]. Following an analogous path, predictive first-principles calculations using density-functional theory (DFT) have become accurate enough to determine the adsorption geometry of these interfaces within experimental error bars irrespective of the adsorbate size using approximate dispersion-inclusive exchange-correlation (XC) functionals[36,43–45]. But whereas benchmark values for the adsorption height of HIOS are available[44], adsorption energies of these interfaces are not fully established beyond the case of small aromatic adsorbates on coinage metal surfaces[36,46].

The state-of-the-art is best exemplified by comparing the adsorption of benzene ($C_6H_6$) and 3,4,9,10-perylene-tetra-carboxylic dianhydride (PTCDA, $C_{24}H_8O_6$) both on the Ag(111) metal surface. Systematic measurements using NIXSW and temperature-programmed desorption (TPD) for benzene on Ag(111) have quantified accurately both the adsorption height and binding strength of a single benzene molecule with a precision of 0.02 Å and 0.05 eV, respectively. Nonlocal DFT calculations including many-body dispersion effects are in remarkable agreement with these measurements within experimental error bars[36]. For the adsorption of PTCDA on Ag(111), accurate values for the adsorption height from experiments and theory exist[39,45,47], but assessment of the adsorption energy has proven to be a challenge. Recent work shows that when using the same parent semilocal PBE exchange-correlation (XC) functional with different dispersion-inclusive methods, the uncertainty in the prediction of the adsorption energy amounts to 1.0 eV[45]. This uncertainty grows to 4 eV when exploring different XC functionals, exceeding by up to two orders of magnitude the chemical accuracy of 0.04 eV. To complicate matters further, an accurate value for the adsorption energy of PTCDA on Ag(111) cannot be determined directly by TPD experiments because, upon heating, PTCDA molecules desorb in fragments due to the strong interaction with the substrate[46]. To validate electronic-structure calculations, two different estimations have been proposed for the adsorption energy of this interface based on a chemically similar but smaller molecule[48,49].

PTCDA can be thought of as a molecule representative of a nanographene system, but with additional functional groups. The disruptive potential of novel materials such as graphene and 2D systems cannot be achieved if challenges due to poor replicability in growth and processing persist along with a limited understanding of the physical and chemical processes underpinning these events in materials science[50]. The comparison shown above demonstrates the challenge, it also shows that quantifying a compound property such as the adsorption energy cannot be achieved by relying on a single methodology.

In this work, we establish a quantitative benchmark for an inorganic-organic interface via two unrelated experimental techniques and nonlocal DFT calculations. Our network of independent methods reinforce and validate each other with the goal of measuring the adsorption energy of the interface formed between PTCDA and Au(111). Such a research design is implicitly characterized by its replicability and let us firmly establish an agreement among methods for the adsorption energy within the uncertainty (precision) of each method.

## Results

**Temperature-programmed desorption experiments**. We start by presenting TPD measurements for PTCDA/Au(111) as a function of PTCDA coverage, applying the so-called complete analysis[51], which gives us access to the binding strength of the interface ($E_{Des}$) by a precise measurement of the desorption rate as a function of temperature at different coverages. Figure 1a shows a series of TPD spectra with different initial coverages for the desorption of PTCDA from the Au(111) surface where three desorption features labeled as $\alpha_1$, $\alpha_2$, and $\alpha_3$ are observed. $\alpha_1$ can be assigned to desorption from the multilayer as it shows characteristics of zero-order desorption and does not saturate whereas $\alpha_2$ can be attributed to desorption from the second layer. The sub- to monolayer desorption is represented by $\alpha_3$ shown in detail in the inset of Fig. 1a. We have used the spectrum corresponding to the saturation of the peak in $\alpha_3$, dark blue curve in the inset of Fig. 1a, to define a monolayer (1.0 ML). The integral of this spectrum works as a reference to determine the coverage of all other TPD spectra. A clear sign for attractive lateral adsorbate-adsorbate and substrate-mediated interactions can be observed in the behavior of the monolayer to sub-monolayer regime as the rising edges of $\alpha_3$ lie on top of each other, while the peak maximum shifts from 694 K for an initial coverage of 0.08 ML to 719 K for a coverage of 1.0 ML.

The desorption follows a first-order process described by the Polanyi-Wigner equation with a rate constant $k = \nu \exp(-E_{Des}/k_B T)$, where the prefactor $\nu$ and the desorption energy $E_{Des}$ characterize the desorption dynamics of the molecule[52,53]. The sticking probability for molecular adsorption is usually close to unity leading to a nonactivated adsorption process in which the adsorption/desorption process is reversible[52,53]. Under these assumptions, $E_{Des}$ is equal to the adsorption energy. It can be determined as a function of coverage from the slope of Arrhenius plots using the complete analysis evaluation routine as exemplary shown for one coverage in the inset of Fig. 1b. The resulting binding energies as a function of PTCDA coverage along with a linear fit to the data (solid line) are displayed in Fig. 1b. The intercept with the $y$-axis gives the desorption energy in the limit of vanishing coverage $E_{Des}(\Theta \to 0 \text{ ML}) = 1.74 \pm 0.10$ eV.

Our TPD analysis effectively extrapolates measurements to the single-molecule level, however, TPD is an experimental technique that probes statistics of adsorbed molecular ensembles. For example, molecular desorption could proceed from more energetically preferred steps or defects and dominate the low-coverage TPD signal[54]. In addition, a rather high desorption temperature of $\approx 700$ K for PTCDA means that a consistent comparison with theory done at 0 K is not unambiguous, since kinetic prefactors can reach values of $10^{24}$ s$^{-1}$ for large flexible molecules[55].

**Scanning probe microscope molecular manipulation experiments**. Complementary to TPD, AFM-based low-temperature single-molecule manipulation experiments represent a novel alternative that can establish a direct link between experiments

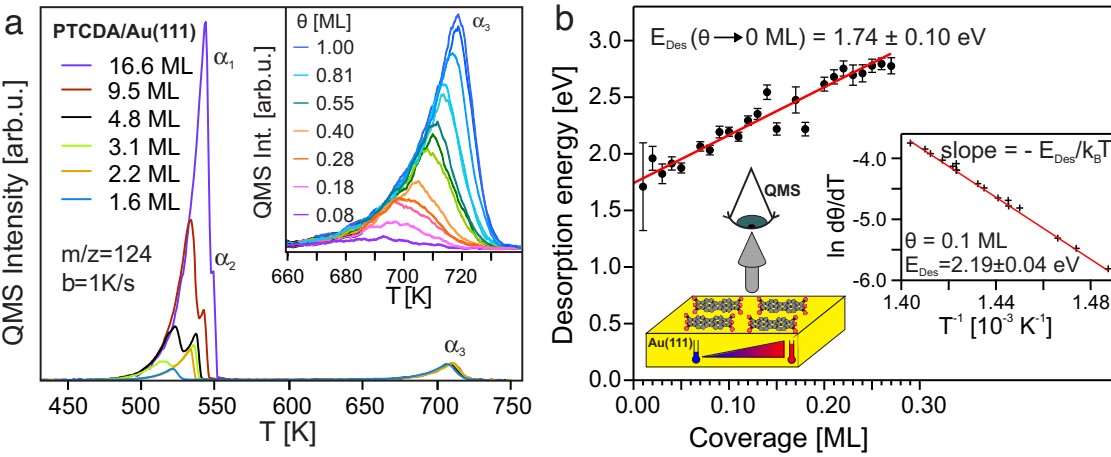

**Fig. 1 Experimental determination of stability for PTCDA/Au(111) system. a** TPD spectra of perylene-tetracarboxylic dianhydride (PTCDA) adsorbed on Au(111) for different initial coverages recorded with a linear heating rate of $\beta = 1$K/s at the fragment mass of 124 amu. The inset shows TPD data of the $\alpha_3$ peak i.e., for sub-monolayer to a monolayer coverages. **b** Desorption energy ($E_{Des}$) as a function of PTCDA coverage determined via the complete analysis. The inset shows the logarithm of the desorption rate (in arbitrary units) as a function of the inverse temperature at an exemplary coverage of 0.1 ML. The slope ($-E_{Des}/k_BT$) of the line yields $E_{Des}$ for that coverage. A linear fit to the data yields the desorption energy in the limit of vanishing coverage ($\Theta \rightarrow 0$ ML; single molecule). The error bars have been produced from the signal to noise ratio of the corresponding TPD data. A visual representation of the TPD experiments is also shown. To record TPD spectra, the samples were heated with a constant heating rate of $\beta = 1$K/s and the desorbing PTCDA molecules were monitored with a quadrupole mass spectrometer (QMS).

and theory at the single-molecule level[42,56,57]. To measure the adsorption energy $E_{ads}$, an isolated molecule is lifted from the surface by the AFM tip, while the gradient $dF_z/dz(z)$ along the surface normal of the force exerted on the molecule is continuously recorded as a function of tip height $z$[56]. This method thus samples the molecule-surface potential $V_{m-s}(\mathbf{x})$, albeit along a complicated trajectory that is defined by the geometries $\mathbf{x}$ the flexible molecule adopts during the lifting process. Consequently, the major challenge of this approach is to find a universal and robust method to obtain $V_{m-s}(\mathbf{x})$ from the measured $dF_z/dz$ data. Here, we present such an approach that solves this inverse problem by forward simulation with a highly flexible atomistic model the parameters of which are fitted to reproduce the experimental data. Most notably, and practically unlike any other experimental technique, the thus obtained potential $V_{m-s}(\mathbf{x})$ yields the molecule-surface interaction for any molecular geometry. We apply our approach to a reference dataset containing 226 individual manipulation experiments[56,57] of PTCDA on Au(111) obtained with a qPlus-type[58] non-contact atomic force / scanning tunneling microscope (NC-AFM/STM) operated at 5 K in ultrahigh vacuum.

The purpose of the parameterization is to enable searching through all possible (reasonable) adsorption potentials. Therefore, the model needs the flexibility to reproduce these potentials such that we can, in principle, find the potential that matches the experimental data best. Simultaneously, the parameter space of the model needs to be compact enough to allow sampling it with reasonable computational effort such that we can actually find the best potential. In our model, we partition the (relative) total energy of the tip-molecule-surface system into contributions from the distortion of the tip (single harmonic potential), from the distortion of the molecule [force field fitted to DFT calculations for gas phase PTCDA and PTCDA/(Au111) (see Fig. S6)], and from the molecule-surface interaction (see Supplementary Methods in the Supplementary Information). The latter is modeled using a set of atom-surface potential functions $V_X(z)$ including terms for chemical attraction, Pauli repulsion, and van der Waals (vdW) interaction (dispersion) which are parameterized separately for each atomic species in PTCDA ($X = $ C,H,O) (see Figs. S1 and

S5). While chemical attraction and Pauli repulsion are well-represented by exponential functions, no such simple representation exists for the short-range vdW interaction which is known to deviate from its asymptotic long-range dependency $V_{vdW}(z) = C_3/(z - z_0)^{-3}$[59] below an atom-surface distance of $z = 4.5$ Å[57]. We address this uncertainty by representing $V_{vdW}(z)$ with a cubic spline that is matched to the asymptotic expression at $z = 4.5$ Å up to its second derivative. Each of the three splines is described by nine parameters, each Pauli-repulsion exponential by two parameters, and the chemical Au-O attraction is modeled by an exponential function with two parameters as well (see Supplementary Methods of the Supplementary Information for details).

To solve the inverse problem, we have simulated lifting curves $dF_z/dz(z)$ (Fig. 2d) for a total of 600,000 randomly parameterized atom-surface potential functions $V_X(z)$ (Fig. 2a, b). These were selected after a plausibility check, from a total of 27 billion samples. In this check, we verify from the potential functions that the adsorption energy at the experimentally determined adsorption height is negative and the estimated adsorption height is within a 0.12 Å interval around the experimental value. Potential energy functions which do not fulfill these criteria are rejected without simulating the lifting process (see also Supplementary Discussion in the Supplementary Information). A given set of functions $V_X(z)$ is considered to be a good approximation of the actual molecule-surface potential if the corresponding simulated $dF_z/dz(z)$ curve matches the measured ones (Fig. 2d), i.e.,

$$\chi^2 = \frac{1}{n}\sum_n (\Delta f_{sim}(z_n) - \Delta f_{exp}(z_n))^2 / \sigma^2_{exp}(z_n) \qquad (1)$$

yields a low value. The variations among different positions of the molecules on the surface with respect to the herringbone reconstruction are smaller than the variations which occur due to different paths of the molecule taken during the repeated lifting-lowering cycles. To minimize the effects which originate from surface corrugation[60] and are not considered in the simulation, we use the average overall 226 experimental curves[56] (Fig. 2d). In this average curve, the effects of different paths of the molecule across

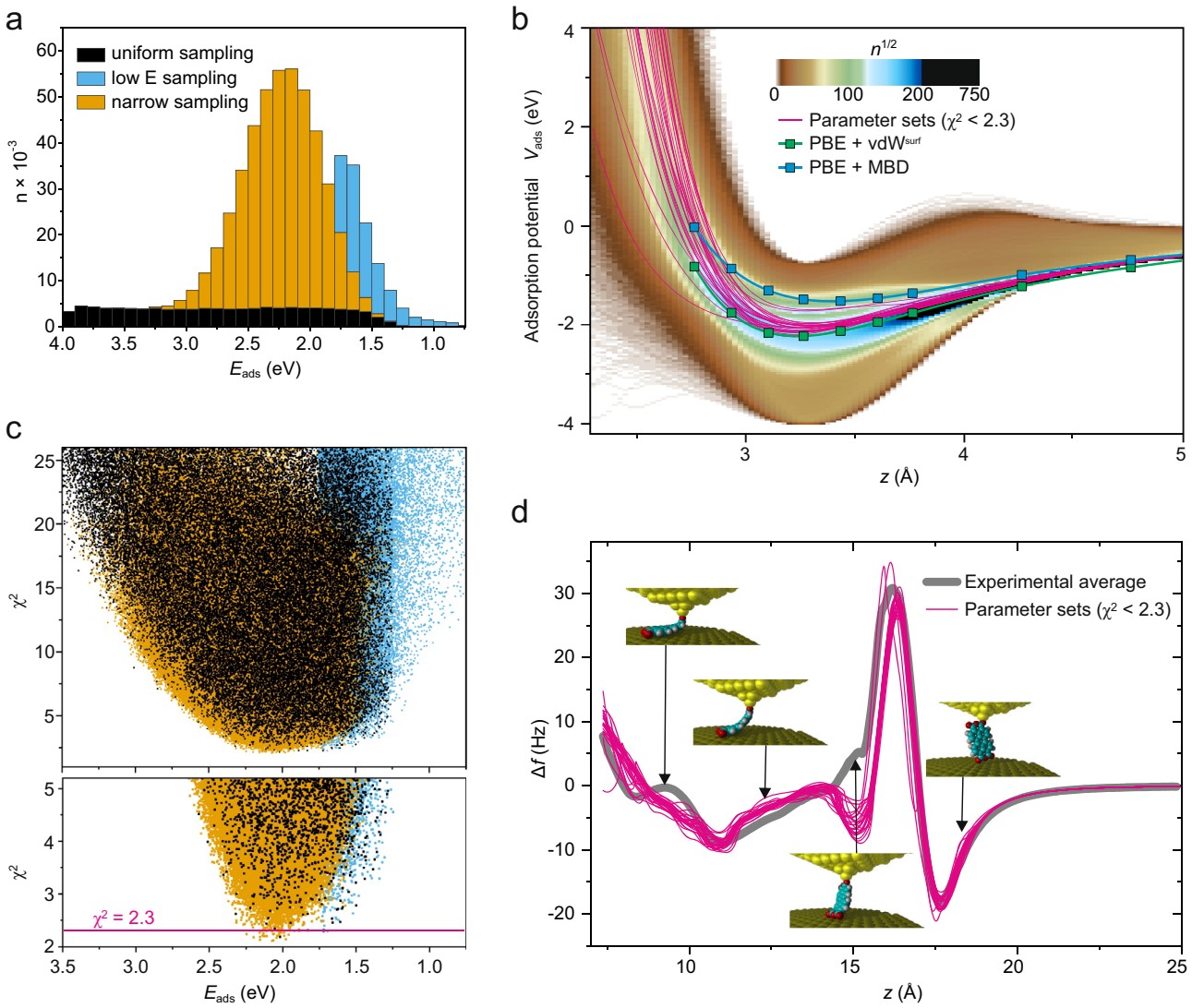

**Fig. 2 Molecule-surface potential fit results. a** Stacked histogram of the number of PE-function samples per 0.1 eV adsorption energy interval. Color coded are three batches of 2 × 100k and 1 × 400k samples. **b** All ≈82k PE-function samples with $\chi^2 \leq 25$ (top) and $\chi^2 \leq 5$ (bottom), color coded according to panel (a). PE-function samples below a threshold of $\chi^2 < 2.3$ (pink line) are plotted in (c) and (d). **c** 2D histogram (bin sizes 0.025 Å × 40 meV) of all molecule-surface potential curves of the three batches. The colour scale shows how often a certain $E(z)$ value was sampled. To emphasize also scarcely sampled regions, we plot $n^{1/2}$ instead of $n$ and put a threshold to black at $n^{1/2} = 200$. We explicitly show the molecule-surface potentials with the best correspondence to the experimental data ($\chi^2 < 2.3$, (d), pink). Single-molecule PE curves calculated with the PBE+vdW$^{surf}$ and PBE+MBD methods are also included as comparison. **d** All simulated $\Delta f(z)$ curves with $\chi^2 < 2.3$ compared to the experimental $\Delta f(z)$ curve (thick gray) which is the average over 226 individual curves[56]. The insets show the conformation of the PTCDA molecule at four different tip heights.

the corrugated surface cancel to a large degree. The only exception occurs at $z = 15$ Å where the molecule is almost vertical and the Au-O corrugation effects therefore maximal and do not cancel out completely. As a consequence the experiment cannot be fully reproduced by the simulation in this region (Fig. 2d). However, this discrepancy has no impact on our conclusions.

In scanning probe microscopy (SPM), the tip morphology is generally not known and the precise strength of the chemical interaction between PTCDA and the tip cannot be explicitly determined. We expect little influence from this fact since the way in which the individual atoms of PTCDA are detached from the surface is unaffected by precise modalities of this bond. Moreover, we account for different tip morphologies insofar as the tip-surface interaction is subtracted from the measured $dF_z/dz(z)$ data before the analysis.

With a total of 35 free parameters for $V_X(z)$, the parameter space of our model cannot be sampled densely such that an

appropriate graphical representation of our results is necessary to estimate the uncertainty related to sparse sampling. Since measuring the adsorption energy of PTCDA on Au(111) is one of our central objectives, we plot the $\chi^2$ values versus $E_{ads}$ in Fig. 2b. When narrowing down the plot to $\chi^2 < 5$, the distribution clearly indicates a value of $E_{ads} = 2.00 \pm 0.25$ eV. In the plots, we show results for three individual sampling runs aimed either at converging the $V_X(z)$ curves irrespective of the huge parameter space (orange and blue in Fig. 2a, b) or at providing a uniform sampling along $E_{ads}$ (black) meant to demonstrate that the convergence to $E_{ads} = 2.00$ eV is *not* the result of a more intense sampling in this energy region.

The best matching $dF_z/dz(z)$ curves ($\chi^2 < 2.3$) are plotted in Fig. 2d. Their mutual similarity proves that our sampling indeed converged to a point where the remaining variations between the simulated curves are smaller than the deviation from the experimental average (gray). This deviation is, hence, of systematic

nature and cannot be eliminated by further searching the parameter space. A likely reason for the most prominent deviation at $z = 15$ Å is the surface corrugation, which has a particularly strong influence at this $z$ value where the molecule is practically vertical[60] such that the corresponding scattering among the 226 experimental curves and thus $\sigma^2_{exp}$ in Eq. (1) is large in the respective region.

Since each parameter set defines the complete molecule-surface potential $V_{m-s}(\mathbf{x})$, we can calculate the adsorption potential curve $V_{ads}(z) = V_{m-s}(\mathbf{x})$ in which $\mathbf{x}$ defines a flat molecule aligned parallel to the surface at distance $z$. The histogram in Fig. 2c shows all 600,000 adsorption potentials obtained from the random sampling, illustrating their high variability. Notably, most $V_{ads}(z)$ curves with very low $E_{ads}$ values show an unphysical energy maximum around $z = 4$ Å. All $V_{ads}(z)$ curves for the best fits (pink) show a very similar behavior and mainly deviate in the region of strong repulsive forces that is barely probed in the molecular manipulation process (see also Fig. S1).

Since our approach can recover the complete adsorption potential $V_{ads}(z)$ from experiment, it offers a unique opportunity to assess theoretical calculations beyond the equilibrium values of adsorption energy and height. Likewise, the deviation between the adsorption energy measured by molecular manipulation ($E_{ads} = 2.00 \pm 0.25$ eV) and by TPD ($E_{ads} = 1.74 \pm 0.10$ eV) asks for further explanation by theory.

**Predictive first-principles modeling of complex inorganic-organic interfaces.** The main challenges in modeling the interaction between PTCDA and Au(111) is the need for a robust description of nonlocal XC effects for this hybrid interface and the explicit modeling of the reconstructed Au(111) surface. This is depicted schematically in single-molecule PE curves calculated with the PBE+vdW$^{surf}$ and PBE+MBD methods shown in Fig. 2c. These results show that the experimentally determined PE curve lies between both models, which differ by the inclusion of many-body effects in the dispersion energy. In fact, these curves only show an approximation to the adsorption energy. Our analysis demonstrates that there are several factors that alter its determination in these complex interfaces. This takes us to our modeling strategy, which is based on identifying two general aspects that influence how to determine the adsorption energy: the limitations of the physical models in standard first-principles calculations and the atomistic structural models. The sequence of steps in this strategy is summarized schematically in Fig. 3a.

The formation of organic-inorganic interfaces such as PTCDA/Au(111) leads to the coupling of the electronic response of each of the components of the interface, a feature that is difficult to capture accurately in state-of-the-art electronic-structure models due to the simultaneous presence of localized and delocalized (free) electronic states. Free-electron screening in metals gives rise to a dynamically screened interaction between ions which effectively reduces the magnitude of dispersion interactions present in the adsorption of organic molecules on metal surfaces. This is a crucial factor to predict the correct atomistic structure and resulting stability of these interfaces[43,44,47,49]. As a first step, we have included this effect by calculating the adsorption energy with the PBE+vdW$^{surf}$ method, yielding an adsorption energy of 2.15 eV for a PTCDA surface density of $\Theta = 1.0$ ML, equivalent to the experimental full monolayer coverage, using the free-standing monolayer in periodic boundary conditions (PBC) as reference state, and 3.06 eV with the single molecule in gas phase as reference state (see Fig. 3b and the Supplementary Discussion in the Supplementary Information for the adsorption energy definitions). The structural model for this calculation is based on a herringbone lateral arrangement of two molecules per unit cell[43,44,61] (see also Fig. S2a). The number of substrate layers,

k-point grid, basis set and their numerical settings, have been chosen to yield adsorption energies converged within 30 meV, which details can be found in the Supplementary Methods of the Supplementary Information (see Figs. S7–S8 and Tables S5–S7).

A PTCDA monolayer is formed via strong intermolecular forces as demonstrated in our TPD experiments (see inset of Fig. 1a). In fact, the difference between our calculations with both reference states is a measure of the intermolecular forces between PTCDA molecules. Clearly, the adsorption energies at $\Theta = 1.0$ ML do not represent the single-molecule limit measured in either of our experiments. For a comparison of the same quantity between theory and experiments, we have calculated the adsorption energy at different PTCDA surface densities $\Theta$, namely 0.60, 0.45, 0.30, and 0.15 ML; using both, the respective free-standing PTCDA surface density at $\Theta$ in PBC and the single molecule in gas phase, as references. This procedure is an effective extrapolation of the adsorption energy to the single molecule given that the limit of low PTCDA surface density for both reference states should be virtually equivalent. The structural models for lower surface densities correspond to a larger unit cell with a three layers slab, in contrast to the five layers slab used for the interface at $\Theta = 1.0$ ML. Figure 4a shows the atomistic models for a monolayer coverage (1.0 ML) and an exemplary PTCDA surface density of 0.45 ML in PBC. Further details of the structural models (see Fig. S2) and the calculation settings can be found in the Supplementary Methods of the Supplementary Information. A summary of the adsorption energies at these surface densities can be found in Table S2 of the Supplementary Information.

Notably, at $\Theta = 0.15$ ML, our calculations show that the difference between both reference states amounts to just 0.04 eV. Taking the average value between both reference states as the equivalent to the limit of low coverage in our calculations, the adsorption energy at the single-molecule level is 2.14 eV with the PBE+vdW$^{surf}$ method (see Fig. S3 and Table S2 for the adsorption energies at all coverages). Based on a comparison between our calculations for $\Theta = 1.0$ ML using three and five layers for the substrate slab, the value of 2.14 eV is underestimated by ~0.08 eV due to the number of substrate layers. Correcting this value by 0.08 eV results in an adsorption energy of 2.22 eV at $\Theta \approx 0.0$ ML. See the Supplementary Discussion in the Supplementary Information for a detailed description.

The PBE+vdW$^{surf}$ method is effectively a pairwise approximation, resulting in an overestimation of the adsorption energy due to the absence of (collective) many-body effects in the dispersion energy[36,49]. To overcome this limitation, we have performed calculations with the PBE+MBD method using the screened polarizability of the DFT+vdW$^{surf}$ method as input for the gold substrate atoms to approximate the effects that the collective behavior of delocalized electrons of the surface have on the dispersion energy. Our calculations at $\Theta = 1.0$ ML show that including many-body effects in the dispersion energy reduces the magnitude of the adsorption energy by approximately 0.70 eV no matter the reference state, whereas calculations with the more recent PBE+MBD-NL functional[62] show a small difference of 10 and 30 meV with respect to the PBD+MBD calculations for both definitions of the adsorption energy, respectively (a summary of the adsorption energies at $\Theta = 1.0$ ML and $\Theta = 0.5$ ML can be found in Table S1 of the Supplementary Information). For comparison, the uncertainty interval of our TPD results is 200 meV and of the SPM-based results 500 meV. These calculations validate the PBE+MBD method for interfaces including metals with screened polarizabilities used as input as the differences fall well inside any experimental error bars (see the Supplementary Discussion and Table S1 of the Supplementary Information for details). Following the same procedure to obtain the limit of a single molecule, many-body

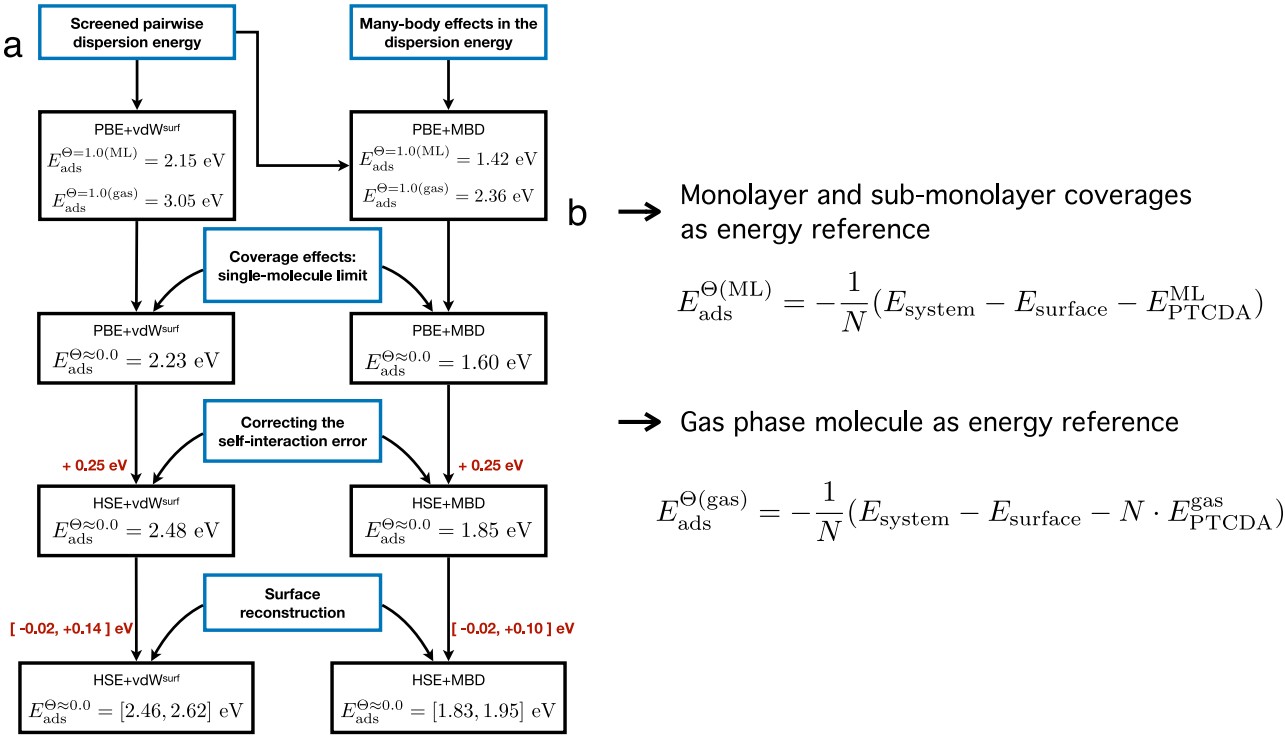

**Fig. 3 Predictive first-principles modeling strategy. a** Modeling strategy based on identifying the factors that determine the quantification of the adsorption energy: (i) free-electron screening, (ii) many-body effects in the dispersion energy, (iii) coverage effects, (iv) influence of the self-interaction error in DFT, and (v) atomistic surface reconstruction. **b** Two definitions for the adsorption energy to study the coverage effects and extrapolate to the single-molecule limit are shown. The definition that takes the energy of an isolated PTCDA molecule in gas phase as reference ($E_{\text{ads}}^{\Theta(\text{gas})}$) includes contributions coming from molecule-metal and molecule-molecule interactions. The definition taking the energy of a free-standing PTCDA surface density in PBC as reference includes only the contributions coming from molecule-metal interactions as the formation energy of a molecular arrangement at surface density $\Theta$ is already included in $E_{\text{ads}}^{\Theta(\text{ML})}$.

effects reduce the energy by 0.63 eV, yielding an adsorption energy of 1.59 eV in the limit of zero coverage.

The next step in our model is to investigate the effect of the self-interaction error (SIE) in DFT which can be found in organic-inorganic interfaces[63–65]. Recent research indicates that an optimal fraction of exact exchange would be closer to zero in the case of strongly coupled organic-inorganic interfaces where strong charge transfer from the adsorbate to the substrate occurs[65]. For PTCDA/Au(111), there is convincing experimental evidence for weak electronic interfacial coupling[66]. In addition, calculations of bulk solids[67] and benzene adsorbed on coinage metals[36] indicate that a fraction of 0.25 exact exchange as formulated in a hybrid XC functional such as HSE[68,69] improves both cohesive properties and adsorption energies when compared to accurate experimental references. By comparing the adsorption energy calculated with dispersion-inclusive PBE and HSE at surface density $\Theta = 1.0$ ML, we have found an increase in the adsorption energy of 0.25 eV due to the SIE regardless of whether vdW interactions include many-body effects or consist of a pairwise approximation. Thus, including a SIE correction via the HSE functional results in an adsorption energy of 2.47 eV and 1.84 eV with the HSE+vdW$^{\text{surf}}$ and HSE+MBD methods at the limit of zero coverage, respectively. A summary of the comparison between adsorption energies using the PBE and HSE XC functionals can be found in Table S3 of the Supplementary Information.

As a final step, we have quantified the influence of the Au(111) surface reconstruction on the adsorption energy in our structural model. We have taken an atomistic model of the $(22 \times \sqrt{3})$ reconstructed Au(111) surface, optimized at the level of PBE[70], to calculate the adsorption energy taking only into consideration the contribution from vdW interactions with both the vdW$^{\text{surf}}$ and

MBD methods. These calculations include 22 planar molecular configurations of a PTCDA molecule on the reconstructed surface at PTCDA surface density $\Theta \approx 0.09$ ML, where the molecule has been placed in each calculation at a distance of 3.19 Å with respect to the averaged position of the reconstructed surface topmost atomic layer (see Figs. 4d, S4a, and S4b). The compression that exists in the top atomic layer of the reconstructured surface leads to a slight buckling of the top-layer gold atoms such that each surface gold atom is located at a different height. As a consequence, the height of the molecule changes depending on its particular position on the surface which results in a variance of the adsorption energy. Thus, the adsorption energy changes within the interval of $[-0.02, 0.14]$ eV with the vdW$^{\text{surf}}$ method and $[-0.02, 0.10]$ eV in the case of the MBD method when the surface reconstruction is included in the model (see Fig. S4c, d, and Table S4 for the adsorption energies; see Fig. S9 for the convergence of the MBD adsorption energies). In addition, an analysis of the influence of stress on the non-reconstructed Au(111) surface and its impact on the adsorption energy can be found in the Supplementary Information; Tables S8 and S9.

## Discussion

As summarized in Fig. 3a, our calculations show that the adsorption energy of a PTCDA molecule on Au(111) lies between 1.82 and 1.94 eV at the HSE+MBD level of theory, which is the most accurate value according to our analysis since it includes all the relevant aspects that influence its quantification. Remarkably, this number is in very good agreement with both measurements of the adsorption energy. The prevailing benchmark of Bz/Ag(111)[36] and this work strongly suggest that the agreement between experiments

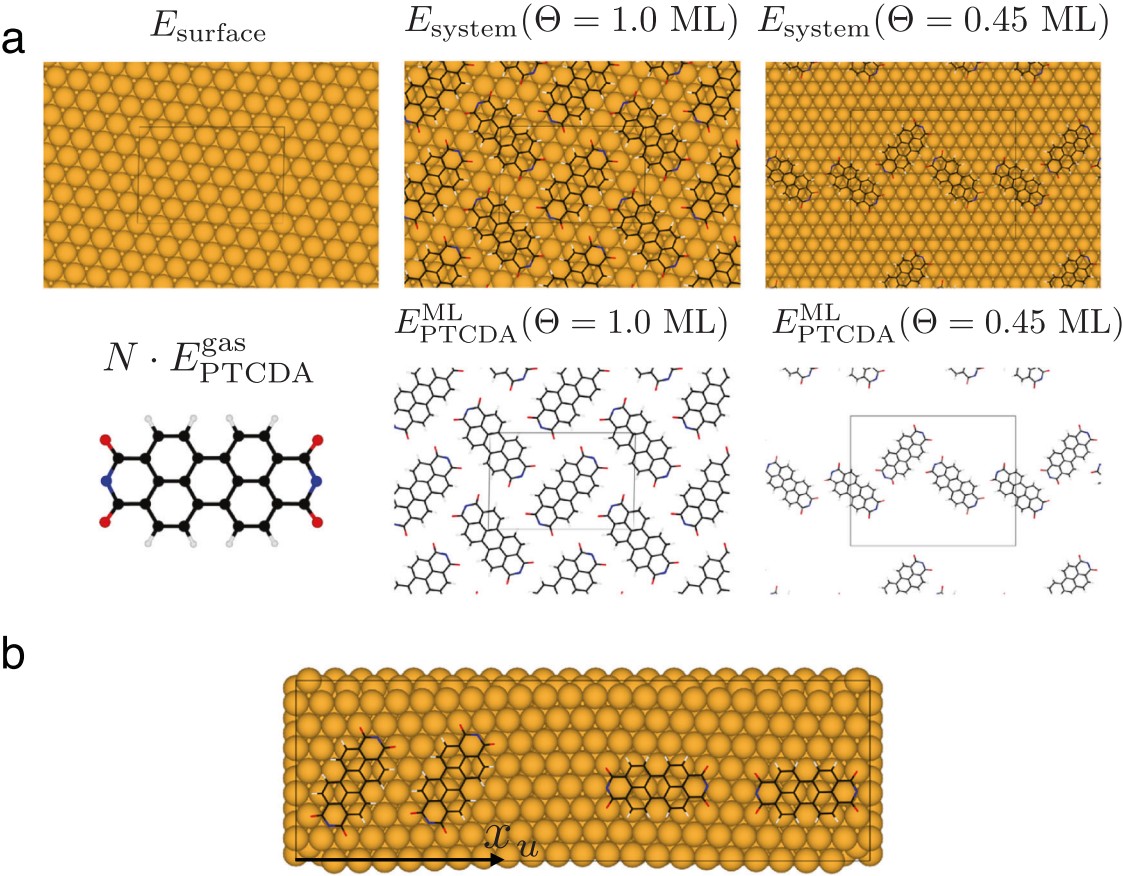

**Fig. 4 Atomistic models for predictive first-principles results. a** Visual representation of the atomistic models of PTCDA/Au(111) at various surface densities. The full monolayer ($\Theta = 1.0$ ML) consists of a herringbone lateral arrangement of two PTCDA molecules per unit cell. An exemplary representation of surface density $\Theta = 0.45$ ML is shown. **b** Atomistic model for a single PTCDA molecule adsorbed on the ($22 \times \sqrt{3}$) reconstructed Au(111) surface. Four exemplary configurations at different adsorption sites out of a total of 22 calculated configurations are shown. These calculations take into account only the contribution from dispersion interactions in the adsorption energy instead of the entire contribution of the XC energy. The full description of all structural adsorption models can be found in the Supplementary Methods of the Supplementary Information.

and the HSE+MBD method is not accidentally right, but rather captures correctly the physics required to model these interfaces. Furthermore, our analysis shows that inclusion of the surface reconstruction can increase the adsorption energy up to 0.12 eV, suggesting a possible explanation for the difference between TPD and AFM lifting experiments. Namely, TPD averages over molecules desorbed from all possible adsorption sites, while AFM measurements correspond to molecules closer to their equilibrium positions. Our calculations also highlight the importance of many-body effects in the dispersion energy leading to a decrease in the adsorption energy. This suggests that the absence of MBD effects in the model to simulate our single-molecule lifting experiments may lead to a slight overestimation of the binding PE curve.

Our unique combined network of first-principles theory and two complementary experimental techniques shows how intricate it is to achieve the same level of atomistic characterization for complex organic-inorganic interfaces as the one established for the case of small aromatic organic molecules. Given that the scientific question of our research have been quantified independently by three unrelated methodologies with an agreement within the uncertainty (precision) of each method, it also demonstrates that our research design has the implicit feature of replicability, as defined by the National Academy of Sciences[71], which has proved to be essential to measure a compound property such as the adsorption energy.

Moreover, the replicability of our network of methods is subtle as the agreement among our results is not perfect. The refinement

in our SPM results and first-principles calculations is, in fact, a consequence of feedback among our methods and identifying missing elements or inaccurate approximations in the physical principles describing our models and experimental interpretations. This demonstrates clearly how non-replicability also plays an important role in the fundamental understanding of system properties that are relevant to materials science. This work has not only established a benchmark quantity, but also exemplifies how implicit replicability in a research design works as a stepping stone in the investigation of materials properties.

## Methods

**TPD experiments**. All TPD experiments were performed under ultrahigh vacuum conditions at a base pressure of $1 \times 10^{-10}$ mbar. The crystals were mounted onto a liquid nitrogen-cooled cryostat, and together with resistive heating a temperature range (measured directly at the substrate via a thermocouple) between 100 and 800 K was achievable and precisely controllable. The Au(111) crystal was prepared by a standard cleaning procedure including $Ar^+$ sputtering and subsequent annealing to 750 K. PTCDA molecules were deposited from an effusion cell held at a temperature of 540 K while the Au(111) surface was kept at room temperature. To record TPD spectra, the samples were heated with a constant heating rate of $\beta = 1$ K/s and the desorbing PTCDA molecules were monitored with a quadrupole mass spectrometer (QMS) at the fragment ion mass of $m/e = 124$. The complete analysis method, which has been applied to analyze the coverage-dependent TPD data, has been described in detail[51].

**Acquisition of SPM data**. Our tip is a Pt-Ir wire cut by an ion beam and prepared by repeated indentation into the Au(111) surface, causing the apex region to be covered with surface material (gold). Only the four carboxylic oxygen atoms at the corners of PTCDA can form a covalent bond to the tip. Since the molecule is

symmetric, these atoms are equivalent and it is therefore irrelevant which of the four atoms is contacted. Each experiment is a sequence (1. Imaging, 2. Contacting, 3. Repeated lifting and lowering, 4. Breaking bond by voltage pulse, 5. Imaging again), such that the azimuthal orientation of PTCDA on the surface and thus the path taken by the molecule during lifting may vary between individual experiments.

**Analysis of force-gradient data**. The experimental curve is compared to the outcome of 600,000 molecular mechanics (MM) simulations of the lifting process. The MM potential functions are defined separately for the intramolecular potential and the molecule-surface potential. The intramolecular potential is described by 33 force field parameters fitted to the DFT energies calculated for 1500 differently distorted PTCDA geometries in gas phase. The molecule-surface potential is described by 35 parameters which are chosen randomly for each simulation. After the parameter generation, a plausibility check is performed, where the random molecule-surface potentials are filtered according to a set of criteria including the predicted adsorption height and plausible energy values for attractive and repulsive potentials. In this step 27 billion random parameter sets were evaluated and the 600,000 reasonable ones were selected for which a full MM simulation was performed. The agreement of all force-gradient curves obtained in this way is then calculated using Eq. 1. All details of the fitting and selection process can be found in the Supplementary Methods of the Supplementary Information.

**DFT general calculation settings**. The structure and the adsorption energy of PTCDA on Au(111) was studied by means of DFT methods. Van der Waals interactions were included as an atom-based pairwise description using the DFT+vdW$^{surf}$ method[43]. Many-body dispersion effects were included via the MBD@rsSCS method[72,73] using the vdW parameters of the DFT+vdW$^{surf}$ scheme as input to approximate the effects of the collective behavior of delocalized electrons within the surface in the dispersion energy. We refer to this method as DFT+MBD, where DFT stands for the XC functional approximation that is being used. A more detailed description can be found in the Supplementary Methods of the Supplementary Information. We performed all calculations using the all-electron/full-potential electronic-structure code FHI-AIMS[74] which uses efficient numerical atom-centered orbitals (NAO) as basis set. Calculations both with *light* and *tight* settings in the FHI-AIMS code were performed, which include *tier 1* standard basis set for Au and *tier 2* basis set for C, H, and O in *tight* settings. The convergence criteria of $10^5$ electrons for the electron density and $10^6$ eV for the total energy of the system were used in all calculations. Relativistic effects were included via the atomic scalar zeroth-order regular approximation[75]. All structural relaxations were performed, including vdW interactions, by establishing a convergence criterion of 0.01 eV Å$^{-1}$ for the maximum final force, without taking into account vdW interactions between metal atoms in order to avoid an artificial relaxation of the surface slab. All MBD calculations were performed as a post-processing step using the many-body dispersion library libmbd[62,76]. An additional calculation of the adsorption energy for the system at full monolayer coverage was performed using the MBD-NL XC functional[62], which unifies nonlocal vdW functionals for polarization and interactions methods for many-body interactions. A detailed description of the adsorption and structural models are to be found in the Supplementary Methods of the Supplementary Information.

## Data availability

The Supplementary Discussion and Supplementary Methods are included in the Supplementary Information that accompanies this paper. The data supporting the TPD and single-molecule AFM experiments is available in the central institutional repository for research data of Forschungszentrum Jülich, Jülich DATA with the identifier https://doi.org/10.26165/JUELICH-DATA/HQXCXC. The data supporting the nonlocal DFT calculations is available in the NOMAD repository with identifier https://doi.org/10.17172/NOMAD/2023.04.30-1 . All relevant data is also available from the authors by request.

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

## Acknowledgements
The authors thank Dr. Felix Hanke for fruitful comments regarding the atomistic model of the reconstructed Au(111) surface. V.G.R. acknowledges support by the North-German Supercomputing Alliance (HLRN). C.W. and H.H.A. acknowledge funding through the European Research Council (ERC-StG 757634 "CM3"). F.M. and P.T. acknowledge the German Research Foundation (DFG) for funding via the collaborative research center SFB 1249 (project B06). F.S.T. acknowledges financial support from the Deutsche Forschungsgemeinschaft through SFB 1083 (project ID 223848855), project A12. A.T. was supported by the European Research Council (ERC-CoG Grant BeStMo) and FNR-CORE Grant "BroadApp" (FNR-CORE C20/MS/14769845).

## Author contributions
F.M., S.S., and P.T. conceived the temperature-programmed desorption experiments and analyzed the results. C.W., H.H.A., and F.S.T. conceived the modeling of the single-molecule manipulation experiments and analyzed the results. V.G.R and A.T. conceived the electronic-structure calculations and analyzed the results. V.G.R, C.W., P.T., F.S.T., and A.T. conceived the final presentation of the results, wrote, contributed to, and edited the manuscript.

## Funding

## Competing interests
The authors declare no competing interests.
