## [Peer Review File · Communications Chemistry]

Reviewers' comments:

Reviewer #1 (Remarks to the Author):

This manuscript titled "Quantifying the Stability of Complex Molecule-Surface Interfaces: Perylene-Tetracarboxylic Dianhydride on Au(111)" is an interesting work about quantitatively measuring the adsorption energy of complex molecule on metal surface. The parameterization-assisted SPM experiment is impressive, which would be potentially generalized to other interfacial adsorption systems. Moreover, the TPD data agree well with the SPM data within the error bar. Therefore, I think that this work deserves the publication in Communications Chemistry. I only have several minor questions for the authors to consider.

1. What kind of tip did the authors use in their SPM study? Is it a tungsten tip ended with a gold atom or other functionalized tips?
2. In figure 2D, there is a dip in the simulated curve while a peak in the experiment curve around $Z=15$ Ångstrom. Are there any explanations for this disagreement?
3. How to make sure the same covalent bond is formed between tip and molecule in each lift? How reproducible are those tip-lifting experiments?
4. There is a typo in line 142. I guess the authors may want to say Au-O attraction.

Reviewer #2 (Remarks to the Author):

This study provides a comprehensive benchmark for quantifying the adsorption energy of PTCDA on Au(111), which poses a replicability challenge and is a well representative case study of organic molecules adsorbed on metal surfaces. The main novelty of the work is the thorough application of three independent techniques (TPD measurements, AFM-based manipulation experiments, and DFT calculations) to assess the agreement among methods. The work nicely extends the authors' previous methodological and applied work on this topic. The theoretical results are performed using state-of-the-art methods capable of simultaneously describing free-electron screening and many-body dispersion effects; this, together with the careful selection of materials models and accounting for some key aspects such as self-interaction error or Au(111) surface reconstruction and stress effects, yields an unprecedented benchmark. Moreover, the theoretical results are compared with comprehensively obtained experimental data, with reasonable good agreement. Importantly, the new approach is expected to help design improved protocols to accurately measure and compute adsorption energies of hybrid organic-inorganic interfaces, which are of paramount relevance for both fundamental and applied research. The topic is therefore timely and relevant for advancing the fundamental understanding of these adsorption systems and publication in Communications Chemistry is justified. Yet, in my opinion, the manuscript requires some clarifications and improvements before it can be accepted:

(1) In Figure 1B a large error (in comparison with that of the other points) can be observed for the desorption energy at the lowest measured coverage. And 2-3 outlier points seem to be present between 0.13-0.2 ML. Could you please comment on the origin and relative importance of these deviations?

(2) For the generation of parametrized potentials for the AFM simulations, please clarify the specific DFT computational details (functional and so on) considered to perform the force field fitting for gas phase PTCDA. And justify why the PBE-vdW^{surf} approach was used for the fitting related to the adsorbed systems instead of the most accurate many-body-dispersion-inclusive MBD or MBD-NL as applied for the first-principles adsorption energy computations.

(3) The DFT adsorption energies are computed at 0 K, which compares poorly with TPD data (obtained at several hundred Kelvin). In principle, it would be possible to approximate the temperature effect by estimating adsorption free energies instead (perhaps neglecting configurational entropy but including the zero-point energy and the $\sum k_B T \ln[1 - \exp(-h\omega_i/k_B T)]$ term). In fact, it could be relevant to include this additional effect of vibrational free energy of the adsorbates in the modelling workflow to see what is its impact to quantify the adsorption energy.

(4) Apparently, the authors sampled the configurational space of different possible adsorption sites only for the single PTCDA molecule model. Please clarify if a similar sampling was also performed for the other considered sub-ML coverages or only one model was computed. If only one model was considered in those cases, please justify the selection.

(5) The choice of the lattice constant of Au for different simulations is a bit confusing. The experimental value is used for computing the adsorption potential energy curve. Then it is mentioned that PBE- and PBE-vdW^{surf}-optimized lattice constants are essentially equivalent, but how they compare with the experimental values? Please clarify also what lattice constant was used for MBD- and HSE-based calculations.

Reviewer #3 (Remarks to the Author):

In this work, authors tried to quantify the stability of complex molecule-Surface interfaces by evaluating the adsorption energy with different methods, including two experimental methods TPD and AFM and the non-local DFT calculations. Three methods mainly analyzed the adsorption of PTCDA on Au (111) surface. The TPD measurement is based on the averages over molecules from all possible adsorption sites, while AFM mainly probes the single molecule closer to its equilibrium. The DFT with the consideration of effects of collective many-body in vdW and self-interaction error should give a reasonable result, by comparing the data from three methods. This work is interesting, and it may be published in Communications Chemistry after the suggestions under,

(1) In Fig. 1b, is the data about desorption energy in y axis that of single PTCDA molecule?

(2) From Fig. 1b, authors obtained the desorption energy 1.74 eV in the limit of vanishing coverage, consistent to the DFT result as discussed in the text. How about the desorption energy at other coverage, such as the coverage of 0.3 ML? Is it also consistent to the results of DFT?

(3) About AFM method, authors stated that they used an atom-surface potential function $V_X(z)$ with a total of 35 free parameters in page 8. May authors give the detailed expression of $V_X(z)$ to discuss the complexity?

(4) In DFT calculations, authors analyzed the adsorption energies under the different coverage, such as $\Theta = 0.15$ ML, 0.30 ML and 1 ML. For each coverage, such as $\Theta = 1$ ML, IS the of arrangement configuration of molecules on a surface in calculations consistent to that in experiments? Could authors give the TEM of $\Theta = 1$ ML?

Response to Reviewers

Response to Reviewer 1

Reviewer's comments:

Summary

This manuscript titled "Quantifying the Stability of Complex Molecule–Surface Interfaces: Perylene-Tetracarboxylic Dianhydride on Au(111)" is an interesting work about quantitatively measuring the adsorption energy of complex molecule on metal surface. The parameterization-assisted SPM experiment is impressive, which would be potentially generalized to other interfacial adsorption systems. Moreover, the TPD data agree well with the SPM data within the error bar. Therefore, I think that this work deserves the publication in Communications Chemistry. I only have several minor questions for the authors to consider.

Authors' reply:

We thank the reviewer for the summary and the positive comments about the paper, as well as the points to improve our manuscript.

Reviewer's query 1:

What kind of tip did the authors use in their SPM study? Is it a tungsten tip ended with a gold atom or other functionalized tips?

Authors' reply:

We now mention the tip in the supplementary information, writing

"Our tip is a Pt-Ir wire cut by an ion beam and prepared by repeated indentation into the Au(111) surface, causing the apex region to be covered with surface material (gold)."

Reviewer's query 2:

In figure 2D, there is a dip in the simulated curve while a peak in the experiment curve around $Z = 15$ Ångstrom. Are there any explanations for this disagreement?

Authors' reply:

The simulation does not consider any surface corrugation and, to account for that, we compare the simulated curves to the averaged experimental curve in which the effects of different paths of the molecule across the corrugated surface cancel to a large degree. This is mentioned in the manuscript. The discrepancy pointed out by the reviewer is located in the region where the molecule is almost vertical and where the corrugation effects are therefore maximal. In this region, they do not cancel out completely and, hence, the experiment cannot be reproduced by a simulation that lacks surface corrugation effects. This is the reason why *none* of the simulated curves in Fig. 2D matches the experiment in this region. However, this discrepancy has no impact on our conclusions. We now mention this in the manuscript, writing:

"In this average curve, the effects of different paths of the molecule across the corrugated surface cancel to a large degree. The only exception occurs at $z = 15$ Å where the molecule is almost vertical and the Au-O corrugation effects therefore maximal and do not cancel out completely. As a consequence the experiment cannot be fully reproduced by the simulation in this region (Fig. 2D). However, this discrepancy has no impact on our conclusions."

Reviewer's query 3:

How to make sure the same covalent bond is formed between tip and molecule in each lift? How reproducible are those tip-lifting experiments?

Authors' reply:

Only the four carboxylic oxygen atoms at the corners of PTCDA can form a covalent bond to the tip. Since the molecule is symmetric, these atoms are equivalent and it is therefore irrelevant which of the four atoms is contacted. Each experiment is a sequence (1. Imaging, 2. Contacting, 3. Repeated lifting and lowering, 4. Breaking bond by voltage pulse, 5. Imaging again), and between individual experiments the azimuthal orientation of PTCDA on the surface may change. This is one

reason why the paths taken by the molecule during lifting vary between experiments (see response to previous question). With a suitable tip apex the experiments are very reproducible, but extensive tip preparation is required to reach this state. The characteristic atomic configuration of a “suitable” tip apex cannot be observed and are therefore not known. We now write in the Supplementary Information:

“Only the four carboxylic oxygen atoms at the corners of PTCDA can form a covalent bond to the tip. Since the molecule is symmetric, these atoms are equivalent and it is therefore irrelevant which of the four atoms is contacted. Each experiment is a sequence (1. Imaging, 2. Contacting, 3. Repeated lifting and lowering, 4. Breaking bond by voltage pulse, 5. Imaging again), such that the azimuthal orientation of PTCDA on the surface and thus the path taken by the molecule during lifting may vary between individual experiments.”

Reviewer’s query 4:

There is a typo in line 142. I guess the authors may want to say Au-O attraction.

Authors’ reply:

We thank the reviewer for pointing this out. We have corrected this in the revised manuscript.

Response to Reviewer 2

Reviewer’s comments:

Summary

This study provides a comprehensive benchmark for quantifying the adsorption energy of PTCDA on Au(111), which poses a replicability challenge and is a well representative case study of organic molecules adsorbed on metal surfaces. The main novelty of the work is the thorough application of three independent techniques (TPD measurements, AFM-based manipulation experiments, and DFT calculations) to assess the agreement among methods. The work nicely extends the authors’ previous methodological and applied work on this topic. The theoretical results are performed using state-of-the-art methods capable of simultaneously describing free-electron screening and many-body dispersion effects; this, together with the careful selection of materials models and accounting for some key aspects such as self-interaction error or Au(111) surface reconstruction and stress effects, yields an unprecedented benchmark. Moreover, the theoretical results are compared with comprehensively obtained experimental data, with reasonable good agreement. Importantly, the new approach is expected to help design improved protocols to accurately measure and compute adsorption energies of hybrid organic-inorganic interfaces, which are of paramount relevance for both fundamental and applied research. The topic is therefore timely and relevant for advancing the fundamental understanding of these adsorption systems and publication in Communications Chemistry is justified. Yet, in my opinion, the manuscript requires some clarifications and improvements before it can be accepted.

Authors’ reply:

We thank the reviewer for the summary and the positive comments about the paper. We also thank the reviewer for the suggestions to clarify and improve the manuscript.

Reviewer’s query 1:

In Figure 1B a large error (in comparison with that of the other points) can be observed for the desorption energy at the lowest measured coverage. And 2-3 outlier points seem to be present between 0.13-0.2 ML. Could you please comment on the origin and relative importance of these deviations.

Authors’ reply:

Indeed the error for the lowest coverage (0.01 ML) is very large. Similarly, errors for 0.02 and 0.03 ML are larger than the others. At very low coverages the TPD data are much more noisy (high signal to noise ratio) compared to the data at higher coverages. This is due to several reasons: (i) There is desorption from other parts of the sample holder, cryostat, etc. during the heating process, leading to the presence of a background signal. (ii) Not all desorbing molecules do reach the ionization region of the QMS and contribute to the QMS signal. (iii) The case of a strong angular-dependent desorption also would lead to a lost in signal intensity. The usage of a so-called “Feulner cup” [1] would solve these problems. However, the relative importance of these errors is low given

that we have enough data points for the linear fit that determines the desorption energy in the limit of vanishing coverage leading to a precision of ± 0.1 eV.

Reviewer's query 2:

For the generation of parametrized potentials for the AFM simulations, please clarify the specific DFT computational details (functional and so on) considered to perform the force field fitting for gas phase PTCDA. And justify why the PBE-vdWsurf approach was used for the fitting related to the adsorbed systems instead of the most accurate many-body-dispersion-inclusive MBD or MBD-NL as applied for the first-principles adsorption energy computations.

Authors' reply:

We now mention the details in the supplementary information, writing:

“For all calculations (gas phase and adsorbed state) the PBE functional and vdW-TS was used, the force threshold was 10^{-2} eV/Angstrom and the SCF cycle convergence criteria for charge density, total energy and sum of eigenvalues were $1E-6$, $1E-6$ and $1E-3$ respectively. The relaxations of the molecule in the adsorbed state were performed in two steps; pre-relaxation with light basis set (BS) followed by relaxation with tight BS. A 4-layer Au(111) slab (area = 23.125×20.027 Angstrom²) was used to simulate the surface and the lattice parameter were converged in advanced. A k-grid of $2 \times 2 \times 1$ was used to sample the BZ and minimum of 12 nm vacuum along z to avoid spurious interactions with periodic cell.”

The respective DFT calculations were performed using FHI-aims version 17 in which MBD calculations were substantially more costly than vdW-TS calculations and MBD-NL was not implemented at all. This is, however, irrelevant, since the aspect limiting the accuracy of the force field approach is not the accuracy of the DFT calculations used for fitting, but the intrinsic limitation of the force field itself. The latter is determined by the used potential energy functions. For performance reasons, they neither include many-body interactions besides angles and torsions, nor non-bonding interactions in our case. Hence, a more sophisticated DFT approach would not improve the FF significantly. Beyond that, creating a new FF would mean to repeat the entire analysis of the experimental data (particularly the extensive Monte-Carlo search), since defining the FF is the first step of the procedure.

Reviewer's query 3:

The DFT adsorption energies are computed at 0 K, which compares poorly with TPD data (obtained at several hundred Kelvin). In principle, it would be possible to approximate the temperature effect by estimating adsorption free energies instead (perhaps neglecting configurational entropy but including the zero-point energy and the $\sum kT \ln[1 - \exp(-h\nu_i/kT)]$ term). In fact, it could be relevant to include this additional effect of vibrational free energy of the adsorbates in the modelling workflow to see what is its impact to quantify the adsorption energy.

Authors' reply:

We agree with the reviewer about the relevance of thermal effects when a comparison between theory and experiment is put forward. In fact, it is clear that entropic effects can become quite important in the quantification of binding free energies of large molecules on surfaces. We must point out however, that it is not our goal to investigate thermal and entropic effects associated to the binding free energy of PTCDA/Au(111) or the comparison between theory and experiments at high temperatures and/or coverage, but to establish a value for the adsorption energy that can be used as a benchmark quantity. It is for this reason that we do not focus on quantifying the free binding energy, but have chosen to quantify the adsorption energy at low temperature and zero coverage.

Furthermore, our interpretation of the TPD experiments is based on transition state theory formulated via the Polanyi-Wigner equation (equations (3) and (4) of the supplementary material). Within this formulation, the energy term E_{des} in the

exponential is not associated with a free energy but represents the activation energy of desorption, which is equal to the adsorption energy under the assumption of a nonactivated process in which the adsorption/desorption process is reversible. Each TPD curve delivers one data point in an

Arrhenius-like representation as stated in the supplementary information. The slope of this representation is the desorption energy E_{Des} as a function of coverage according to equation (4) in the supplementary material. As a consequence, only a correction of up to $k_B T_{\text{Des}}/2$ must be added to the final adsorption energy due to the gas impingement rate. This amounts to approximately 0.02 eV, resulting in an adsorption energy of approximately 1.76 ± 0.10 eV extrapolated at low temperature according to the TPD experiments. In this regard, our experimental setup is not designed to measure free adsorption energies but to focus on the adsorption energy of a single molecule, where the entropic effects are not large enough to substantially modify the comparison between theory and experiment.

Independently of these facts, it is nevertheless important to add that that entropic effects in large molecules become important when the molecule is highly flexible. In organic molecules, this happens mostly with the presence of single bonds that allow chemical functional groups or backbones in large molecules to rotate and wiggle due to thermal effects. An example of this can be found in the adsorption of a single azobenzene molecule on Ag(111) [2] or the adsorption of single oligophenyl molecules on amorphous silica [3]. In contrast, PTCDA is not composed of single carbon bonds that can lead to flexibility, but a structure that is closer to nanographene, making it much stiffer than oligomers or oligophenyl molecules.

Because of these reasons, we do not think that entropic effects modify substantially the discussion of our results or the determination of the adsorption energy. We have added this discussion to the supplementary material under the section *“Temperature programmed desorption measurements”*.

Reviewer’s query 4:

Apparently, the authors sampled the configurational space of different possible adsorption sites only for the single PTCDA molecule model. Please clarify if a similar sampling was also performed for the other considered sub-ML coverages or only one model was computed. If only one model was considered in those cases, please justify the selection.

Authors’ reply:

Contrary to the case of a single molecule on the reconstructed surface, we have not performed a sampling of different structures corresponding to sub-ML coverages. The structural models considered for coverages of 0.60, 0.45, 0.30, and 0.15 ML are shown in Fig. S2 of the supplementary material. The justification of this is that our objective is to obtain an accurate quantification of the adsorption energy of a single molecule (limit of zero coverage). The purpose of calculating energies at molecule surface densities lower than 1.0 ML is to generate a systematic procedure that can be used as a base to extrapolate the adsorption energy to the limit of zero coverage. This convergence is achieved when comparing the adsorption energy at 0.15 ML using both energy references given by equations (1) and (2) of the supplementary material. The convergence can also be observed in Fig. S3 of the supplementary material. This can also be found in the subsection *‘Coverage effects: extrapolation to the single-molecule limit’* of the supplementary material. The quantification of the adsorption energy of the interface at different molecule surface densities is not a straightforward task that therefore falls beyond the scope of this work.

Reviewer’s query 5:

The choice of the lattice constant of Au for different simulations is a bit confusing. The experimental value is used for computing the adsorption potential energy curve. Then it is mentioned that PBE- and PBE-vdWsurf-optimized lattice constants are essentially equivalent, but how they compare with the experimental values? Please clarify also what lattice constant was used for MBD- and HSE-based calculations.

Authors’ reply:

We thank the reviewer for pointing this out, the choice of lattice constant for different calculations is not clearly stated in its entirety of the supplementary information of the manuscript. We have only used two different lattice constants for all the calculations we report in our work. As the reviewer mentions, the experimental lattice constant for Au was used to build the slab corresponding to the PBE+vdW^{surf} and PBE+MBD adsorption potential energy curves shown in Figure 2C of the manuscript. For all other calculations, including those with MBD or HSE-based, the PBE lattice constant was used to build the surface slab.

We have taken the lattice constant values from a previous work [4], corresponding to 4.062 and 4.159 Å for the experimental and PBE values respectively whereas the PBE+vdW^{surf} value is 4.163 Å. Therefore, a small difference of 0.004 Å exists between the PBE and PBE+vdW^{surf} values. The PBE+vdW^{surf} simulates to some extent the screening effect by the metallic bulk electrons. For the vdW energy inside the metallic bulk, this method rather overestimates the interactions between metallic electrons, given the fact that these are already described accurately by the PBE functional since PBE is reduced to the local density approximation (LDA) for homogeneous electron densities [4]. It is because of this reason that we have chosen the PBE lattice constant to generate the surface slab in our calculations.

With respect to the potential energy (PE) curve, a previous calculation [5] of the same curve using the PBE lattice constant used to generate the slab have shown a difference of 0.05 eV in the adsorption energy. This difference may not be negligible itself in the PE curve as we state in the manuscript that this curve works only as an approximation. The goal of calculating the PE curve is not that of accuracy but to illustrate the fact that complexity in the determination of the adsorption energy in these complex interfaces. In this particular case, we can show clearly the effect of including many-body effects in the dispersion energy.

Following the comments of the reviewer, we have added a small subsection in the supplementary material regarding the selection of the lattice constant for our calculations, stating it clearly also in the subsections corresponding to the MBD and HSE-based calculations:

“Lattice constant. The experimental lattice constant for Au was used to build the slab corresponding to the PBE+vdW^{surf} and PBE+MBD adsorption potential energy curves shown in Figure 2C of the manuscript. For all other calculations, including single point calculations with the MBD method and the HSE XC functional, the PBE lattice constant was used to build the surface slab. We have taken these values from previous work (37), corresponding to 4.062 and 4.159 Å for the experimental and PBE values respectively. According to the same work, the PBE+vdW^{surf} lattice constant is 4.163 Å, amounting to a difference of 0.004 Å between PBE and PBE+vdW^{surf}. The PBE+vdW^{surf} method simulates to some extent the screening effect by the metallic bulk electrons, but for the vdW energy inside the metallic bulk, this method rather overestimates the interactions between metallic electrons given that these are already described accurately by the PBE functional since PBE is reduced to the local density approximation (LDA) for homogeneous electron densities (37). It is because of this reason that we have chosen the PBE lattice constant to generate the surface slab in our calculations.”

Response to Reviewer 3

Reviewer’s comments:

Summary

In this work, authors tried to quantify the stability of complex molecule-Surface interfaces by evaluating the adsorption energy with different methods, including two experimental methods TPD and AFM and the non-local DFT calculations. Three methods mainly analyzed the adsorption of PTCDAs on Au (111) surface. The TPD measurement is based on the averages over molecules from all possible adsorption sites, while AFM mainly probes the single molecule closer to its equilibrium. The DFT with the consideration of effects of collective many-body in vdW and self-interaction error should give a reasonable result, by comparing the data from three methods. This work is interesting, and it may be published in Communications Chemistry after the suggestions under.

Authors' reply:

We thank the reviewer for the summary and the positive comments about the paper. We also thank the reviewer for the suggestions to improve the manuscript.

Reviewer's query 1:

In Fig. 1b, is the data about desorption energy in y axis that of single PTCD A molecule?

Authors' reply:

Yes, the reviewer is correct in this query. We state in the manuscript, page 5 line 98-100: "The intercept with the y-axis gives the desorption energy in the limit of vanishing coverage $E_{\text{Des}}(\Theta \rightarrow 0 \text{ ML}) = 1.74 \pm 0.10 \text{ eV}$. To be more clear, we have now included the following in the caption of Fig. 1:

"A linear fit to the data yields the desorption energy in the limit of vanishing coverage ($\Theta \rightarrow 0 \text{ ML}$; single molecule)."

Reviewer's query 2:

From Fig. 1b, authors obtained the desorption energy 1.74 eV in the limit of vanishing coverage, consistent to the DFT result as discussed in the text. How about the desorption energy at other coverage, such as the coverage of 0.3 ML? Is it also consistent to the results of DFT?

Authors' reply:

We cannot compare the coverages used for the calculations with the measured one, since the adsorption structures between theory and experiment at these coverages are completely different. In the low-coverage regime ($< 1 \text{ ML}$) PTCD A/Au(111) will form islands due to attractive interaction between the molecules, which results in an increase of the desorption temperature with rising coverage (see Fig. 1, inset). For example, the experimental value for 0.3 ML is 3.01 eV whereas the value given by theory is around 1.62 eV at the PBE+MBD level of theory using the gas phase molecule as reference. These numbers are not comparable due to the difficulty of reproducing, from the theory perspective, all the conditions at which the TPD experiments take place. The purpose of calculating energies at molecule surface densities lower than 1.0 ML is to generate a systematic procedure that can be used as a base to extrapolate the adsorption energy to the limit of zero coverage from the theory perspective. The convergence is achieved when comparing the adsorption energy at 0.15 ML using both energy references given by equations (1) and (2) of the supplementary material. The convergence can also be observed in Fig. S3 of the supplementary material. This can also be found in the subsection '*Coverage effects: extrapolation to the single-molecule limit*' of the supplementary material. It is just at the single-molecule level that a comparison between theory and experiment is possible. The quantification of the adsorption energy of the interface at different molecule surface densities and its comparison to experiments is not a straightforward task that therefore falls beyond the scope of this work.

Reviewer's query 3:

About AFM method, authors stated that they used an atom-surface potential function $V_X(z)$ with a total of 35 free parameters in page 8. May authors give the detailed expression of $V_X(z)$ to discuss the complexity?

Authors' reply:

The vdW part of the molecule-surface potential are cubic splines:

$$E_{\text{vdW}} = (A_{X,n} \times z^3 + B_{X,n} \times z^2 + C_{X,n} \times z + D_{X,n}),$$

where $A_{X,n}$, $B_{X,n}$, $C_{X,n}$, and $D_{X,n}$ change their values in each of the $n = 10$ z-intervals of the spline and for each atomic species $X = \text{H, C, O}$. The Pauli repulsion is modeled by an exponential

$$E_{\text{Pauli}} = \exp(E_X - F_X \times z),$$

with $X = \text{H, C, O}$ and the covalent interaction between O atoms and the surface is likewise modeled by an exponential

$$E_{\text{Chem}} = -\exp(G - H \times z).$$

The complete atom-surface potential function for H and C atoms is thus

$$E_{\text{vdW}} = (A_{X,n} \times z^3 + B_{X,n} \times z^2 + C_{X,n} \times z + D_{X,n}) + \exp(E_X - F_X \times z),$$

with $X = \text{H, C, and}$

$$E_{\text{vdW}} = (A_{O,n} \times z^3 + B_{O,n} \times z^2 + C_{O,n} \times z + D_{O,n}) + \exp(E_O - F_O \times z) - \exp(G - H \times z)$$

for O atoms.

Since the spline segments are matched up to the second derivative at the points where they are connected, and since the first segment (at $z = 4.5 \text{ \AA}$) matches the asymptotic expression $E_{\text{vdW}} = C_3/(z-z_0)^3$, there are only nine *free* parameters per spline, leading to a total of 27 spline parameters. With 6 parameters for the Pauli repulsion and two for the O-Au attraction, this amounts to a total of 35 parameters. We now mention this in the Supplemental Material.

Reviewer's query 4:

In DFT calculations, authors analyzed the adsorption energies under the different coverage, such as $\Theta=0.15 \text{ ML}$, 0.30 ML and 1 ML . For each coverage, such as $\Theta = 1 \text{ ML}$, IS the of arrangement configuration of molecules on a surface in calculations consistent to that in experiments? Could authors give the TEM of $\Theta=1 \text{ ML}$?

Authors' reply:

The reviewer raises up an interesting query regarding the structures of the sub-monolayer coverages and the comparison of these between theory and experiments. We have partially addressed this above in the reviewer's query 2. Contrary to the case of a single molecule on the reconstructed surface, we have not performed a sampling of different structures corresponding to sub-monolayer coverages. The structural models considered for the calculations corresponding to 0.60 , 0.45 , 0.30 , and 0.15 ML are shown in Fig. S2 of the supplementary material. In this regard, the configuration of the molecules are not consistent or based in experiments. We emphasize that the justification of this is that our objective is to obtain an accurate quantification of the adsorption energy of a single molecule (limit of zero coverage) and, as we have explained above, the quantification and comparison of adsorption energies at different sub-monolayer coverages falls beyond the scope of this work. This is also mainly because a direct comparison between theory and experiment at these coverages is very difficult to achieve and reproduce due to the conditions at which the TPD experiments take place.

For the case of $\Theta=1.0 \text{ ML}$, our base model is briefly discussed in subsection "*Monolayer coverage model and additional calculation settings*" of the supplementary information. PTCDA does not form commensurate monolayer in the Au(111) surface but rather exhibits a very close situation to a point-on-line growth on the reconstructed surface, which is difficult to reproduce with state-of-the-art modeling. Our base model is based on the experimental structure observed for the case of PTCA on the Ag(111) surface, the details can be found in the supplementary material. This is another reason behind our focus on quantifying the adsorption energy at the level of the single molecule. Our entire research setup focuses on establishing an accurate benchmark for the adsorption of a single PTCDA molecule given the fact that establishing energies and structural features at the sub-monolayer and monolayer coverages would not be a task that could be achieved within the time window of this research project.

References

- [1] P. Feulner, D. Menzel. Simple ways to improve “flash desorption” measurements from single crystal surfaces. *Journal of Vacuum Science and Technology* **17**, 662 (1980).
- [2] R. J. Maurer, W. Liu, I. Poltavsky *et al.* Thermal and Electronic Fluctuations of Flexible Adsorbed Molecules: Azobenzene on Ag(111), *Physical Review Letters* **116**, 146101 (2016).
- [3] M. Miletic, K. Palczynski, and J. Dzubiella. Quantifying entropic barriers in single-molecule surface diffusion, *The Journal of Chemical Physics* **153**, 164713 (2020).
- [4] W. Liu, V. G. Ruiz, G. Zhang *et al.* Structure and energetics of benzene adsorbed on transition-metal surfaces: density-functional theory with van der Waals interactions including collective substrate response, *New Journal of Physics* **15**, 053046 (2013).
- [5] V. G. Ruiz, PhD. thesis (Fritz-Haber-Institut der MPG) (2016), available at https://th.fhi.mpg.de/site/uploads/Publications/RuizLopez-Thesis-Final_20160721.pdf.

REVIEWERS' COMMENTS:

Reviewer #1 (Remarks to the Author):

[Editorial note: The reviewer has not left further comments for the authors.]

Reviewer #2 (Remarks to the Author):

The authors have satisfactorily addressed previous concerns and, in my opinion, the manuscript is now publishable in Communications Chemistry.

Reviewer #3 (Remarks to the Author):

[Editorial note: The reviewer has not left further comments for the authors.]